# Induction of Hepatitis E Virus Anti-ORF3 Antibodies from Systemic Administration of a Muscle-Specific Adeno-Associated Virus (AAV) Vector

**DOI:** 10.3390/v14020266

**Published:** 2022-01-27

**Authors:** Lars Maurer, Jihad El Andari, Kleopatra Rapti, Laura Spreyer, Eike Steinmann, Dirk Grimm, Viet Loan Dao Thi

**Affiliations:** 1Schaller Research Group, Department of Infectious Diseases, Virology, University Hospital Heidelberg, Center for Integrative Infectious Diseases Research (CIID), 61920 Heidelberg, Germany; lars.maurer01@stud.uni-heidelberg.de (L.M.); Laura.Spreyer@stud.uni-heidelberg.de (L.S.); 2Department of Infectious Diseases, Virology, University Hospital Heidelberg, Cluster of Excellence CellNetworks, BioQuant, Center for Integrative Infectious Diseases Research (CIID), 69120 Heidelberg, Germany; josef.elandari@gmail.com (J.E.A.); kleopatra.rapti@bioquant.uni-heidelberg.de (K.R.); 3Department of Molecular and Medical Virology, Ruhr-University Bochum, 44801 Bochum, Germany; eike.steinmann@ruhr-uni-bochum.de; 4German Center for Infection Research (DZIF), External Partner Site Bochum, 44801 Bochum, Germany; 5German Center for Infection Research (DZIF), Partner Site Heidelberg, 69120 Heidelberg, Germany; 6German Center for Cardiovascular Research (DZHK), Partner Site Heidelberg, 69120 Heidelberg, Germany

**Keywords:** adeno-associated virus, AAV, hepatitis E virus, HEV, vector-based vaccine, neutralisation

## Abstract

The hepatitis E virus (HEV) is a major global health problem, leading to large outbreaks in the developing world and chronic infections in the developed world. HEV is a non-enveloped virus, which circulates in the blood in a quasi-enveloped form. The quasi-envelope protects HEV particles from neutralising anti-capsid antibodies in the serum; however, most vaccine approaches are designed to induce an immune response against the HEV capsid. In this study, we explored systemic in vivo administration of a novel synthetic and myotropic Adeno-associated virus vector (AAVMYO3) to express the small HEV phosphoprotein ORF3 (found on quasi-enveloped HEV) in the musculature of mice, resulting in the robust and dose-dependent formation of anti-ORF3 antibodies. Neutralisation assays using the serum of ORF3 AAV-transduced mice showed a modest inhibitory effect on the infection of quasi-enveloped HEV in vivo, comparable to previously characterised anti-ORF3 antibodies used as a control. The novel AAVMYO3 capsid used in this study can serve as a versatile platform for the continued development of vector-based vaccines against HEV and other infectious agents, which could complement traditional vaccines akin to the current positive experience with SARS-CoV-2.

## 1. Introduction

With up to 20 million infections and 3.3 million symptomatic cases each year, the hepatitis E virus (HEV) is one of the most common causes of acute hepatitis worldwide [1,2]. HEV infections are mostly self-limiting but can progress to chronicity in immunocompromised patients [1,3]. In addition, high mortality rates in pregnant women were reported [1,4]. Despite posing a global health problem, specific anti-HEV treatments remain urgently needed, and only two HEV vaccine candidates have been evaluated in clinical trials thus far (reviewed in [5]). Both vaccines have demonstrated high efficacy in preventing acute HEV infection, but only HEV 239 (Hecolin^®^) was further developed, and is currently only available in China and a few other countries [6].

HEV is classified in its own *Hepeviridae* family (reviewed in the Ref. [7]). The four main genotypes (GT) 1 to 4 infecting humans belong to the *Orthohepevirus A* species [7]. GTs 1 and 2 are restricted to humans, mainly transmitted via the faecal–oral route, and are highly prevalent in countries of East and South Asia [7]. GTs 3 and 4 can infect a broad range of hosts, including pigs, rabbits, and deer [8]. They are mainly transmitted to humans zoonotically by the consumption of undercooked meat products and are responsible for the majority of HEV infections in Europe and North America [7]. In addition, human-to-human transmission can occur through blood transfusions [9]. Other genotypes, such as GT7 [10] and GT1 of the *Orthohepevirus* C species [11], have also recently been found to infect humans.

HEV is a non-enveloped (nHEV), positive-strand RNA virus (reviewed in the Ref. [12]). Its 7.2 kb genome contains three open-reading frames (ORF1-3) (Figure 1A). ORF1 encodes the domains mediating genome replication; ORF2, the capsid protein; and ORF3, a small phosphoprotein that is critical for HEV secretion [12]. HEV GT1 viruses express an additional ORF4, which can enhance the replication of both GT1 [13] and GT3 [14] viruses when expressed *in trans*.

ORF3 interacts with the protein tumor susceptibility gene 101 (TSG101) of the endosomal sorting complexes required for transport (ESCRT) via its PSAP motif, which is critical for HEV budding into multivesicular bodies [15]. Despite the lack of viral glycoproteins, HEV gains a membranous, host-derived envelope during the process of secretion from cells [16], similar to the hepatitis A virus [17]. In this form, the virus circulates in the blood and is called a quasi-enveloped HEV (eHEV), bearing both ORF2 and ORF3 (Figure 1B) [12,18,19]. The quasi-envelope is removed via bile salts [20], yielding nHEV particles which are shed into faeces for transmission to another host [12]. The envelope confers protection of HEV particles from neutralising anti-ORF2 antibodies in the serum [21]. eHEV particles in the blood are potentially responsible for intrahepatic spread, as well as dissemination to other organs [22], as evidenced by extrahepatic manifestations in chronic HEV patients [23]. In agreement with its essential role in mediating HEV release, ORF3 is only found on eHEV, but not on nHEV particles [24].

To date, most vaccine approaches aiming at preventing HEV infections have been designed to induce an immune response against capsid ORF2, but not ORF3. However, a recent study showed that HEV may cross the intestinal barrier through active infection [25]. The observation that progenies released from enterocytes are quasi-enveloped eHEV particles suggests that this is the primary form reaching the liver and establishing infection. In addition, as mentioned before, transmission through blood transfusions, in which mostly ORF3-displaying eHEV particles are circulating, has been reported. Finally, *in vitro* studies have shown that anti-ORF3 antibodies (Abs) can capture viral particles from serum samples of HEV patients [21] or from supernatants of HEV-infected cells [15,26,27], and that they are able to partially neutralise eHEV infections *in vitro* [21,24].

Altogether, this encouraged us to re-evaluate whether eHEV particles and, accordingly, ORF3 are useful targets for an anti-HEV vaccine, in addition to existing anti-ORF2 vaccines. To this end, we harnessed the suitability of the Adeno-associated virus (AAV), a widely used scaffold for therapeutic gene delivery in humans, to engineer a vector-based vaccine. AAV is a member of the *Parvoviridae* family and composed of a single-stranded DNA genome packaged into a small, non-enveloped icosahedral capsid (reviewed in [28]). Advantages of AAVs as a gene delivery vector in humans are their lack of pathogenicity and replication deficiency in the absence of a helper virus (typically Adenovirus) (reviewed in the Refs. [29,30]). The majority of recombinant (r)AAV genomes persist as episomes in the nucleus and the rate of AAV integration in the host genome is low, which adds to the favourable AAV safety profile in patients (reviewed in the Ref. [28]). Moreover, the AAV genome and capsid are both highly amenable to molecular engineering and directed evolution, which facilitates the creation of designer vectors with optimal tissue or cell specificities, desired levels of transduction, and/or low reactivities with neutralising anti-AAV antibodies [28,30,31,32,33,34,35]. The latter is particularly beneficial when using AAV as a vaccine platform, as capsid-swapping (also called pseudotyping) allows for repeated vector administration and, accordingly, booster strategies [36].

In general, the hallmark of the idea to use viral vectors as a vaccine is delivery of a nucleic acid (DNA, in the case of AAV) encoding and expressing a selected viral antigen in the recipient’s cells. The expression of the heterologous protein can then induce a humoral and cellular host immune response against the pathogen from which the foreign antigen is derived [37]. Several studies have shown that the AAV-mediated expression of non-self-antigens can yield strong and sustained antibody responses, most likely due to the high and long-lasting transgene expression in vivo [38,39,40] (reviewed in the Ref. [36]). Moreover, many AAV capsid variants are stable under various physical conditions, including a wide temperature range, and are amenable to lyophilisation [38]. Together with the constant advances in AAV manufacturing and the high versatility of this vector system, this makes AAV an interesting and promising candidate for global vaccination campaigns. Its great potential is exemplified by an encouraging recent preclinical study in which AAVs were engineered to express the SARS-CoV-2 spike (S) antigen and shown to induce a robust anti-viral immune response in non-human primates [38]. This raises hopes that AAV-based anti-SARS-CoV2 vaccines could soon complement those based on several other viral vectors that have already been authorised for use in humans conferring efficient protection against SARS-CoV2 infection [41].

Here, we pursued the two related aims to (1) re-evaluate the potential benefit of anti-ORF3 antibodies as an anti-HEV vaccine, and to (2) assess the capacity of an optimized AAV vector for direct induction of these antibodies in vivo. Therefore, we combined rapidly expressing self-complementary [42] AAV vector genomes with a novel chimeric and myotropic capsid called AAVMYO3 that yields efficient and specific transgene expression in the entire musculature following peripheral administration. Following intravenous delivery into mice and expression of ORF3, the potential of the induced anti-ORF3 antibodies to neutralise eHEV particles was studied in vitro and found to match that of previously described anti-ORF3 antibodies.

## 2. Materials and Methods

### 2.1. Plasmids and Cells

A plasmid encoding HEV GT3 Kernow-C1/p6 (GenBank accession No: JQ679013) and human hepatoma S10-3 cells were kindly provided by Suzanne Emerson (NIH). The AAV helper plasmid from which the AAVMYO3 helper plasmid was derived (El Andari et al., submitted) as well as the adenoviral helper plasmid have been reported previously [44]. S10-3 and HEK-293 cells (ATCC CRL-1573; LGC Standards GmbH, Wesel, Germany) were grown in Dulbecco’s Modified Eagle’s Medium (Gibco, Carlsbad, CA, USA) supplemented with 10% fetal bovine serum (FBS; Merck, Darmstadt, Germany) and 1% penicillin-streptomycin (Gibco, Carlsbad, CA, USA).

### 2.2. AAV Production and Titration

The HEV ORF3 gene from the HEV GT3 Kernow-C1/p6 plasmid was cloned into a self-complementary AAV vector (pscAAV-CMV-EYFP-BGHpolyA [45]), via overlap PCR and using the NotI and BsrGI restriction sites. In the resulting construct, the ORF3 transgene was under the control of a cytomegalovirus (CMV) promoter and enhancer, with an SV40 intron and a bovine growth hormone (BGH) polyA signal, and flanked by inverted terminal repeats (ITRs) from AAV2 and AAV4. Recombinant AAVs were produced as described before [46]. In brief, adherent HEK-293 cells grown in 15 cm dishes (Nunc/Thermo Fisher Scientific, Waltham, MA, USA) were transfected with 14.6 µg each per plate of the recombinant AAV vector plasmid (encoding the ORF3 or *yfp* transgene), the AAV helper plasmid encoding AAV2 *rep* and the synthetic capsid gene MYO3, and an adenoviral helper plasmid (Figure 1C). The AAVs were harvested from the cell pellet, purified by iodixanol density gradient centrifugation (15, 25, 40, 60% iodixanol) and then buffer-exchanged to PBS and concentrated through Amicon Ultra-15 (Merck, Darmstadt, Germany) columns (100 kDa). AAV genomes were titrated by RT-qPCR on a Rotor Gene 6000 cycler (QIAGEN, Hilden, Germany) as described before [46], using probes and primers directed at the CMV enhancer element (listed in Appendix A). AAV vector titers typically exceeded 5 × 10^12^ genome copies/mL.

### 2.3. Animals

Six-week-old female inbred BALB/c mice (Charles River, Sulzfeld, Germany) were injected intravenously into the tail vein with 1 × 10^11^ or 1 × 10^12^ viral genomes (vg) per mouse in 100 μL PBS. Before injection and every two weeks thereafter, 60–80 μL blood was taken by facial vein puncture, and the serum was stored at −80 °C. After 10 weeks, all mice were euthanised by cardiac puncture under intraperitoneally administered Ketamin/Xylazine anaesthesia (120 mg/kg Ketamin and 16 mg/kg Xylazine). Liver, spleen, the quadriceps femoris muscle, and 600–800 μL blood were harvested. The organs were submerged in RNAlater (Thermo Fisher Scientific, Waltham, MA, USA) and stored at −20 °C, and the serum was stored at −80 °C.

### 2.4. Real-Time Quantitative RT-PCR

Total RNA from mouse tissues was isolated using the Qiagen AllPrep DNA/RNA purification kit (QIAGEN, Hilden, Germany) according to the manufacturer’s instructions. In brief, 10–20 mg of each tissue was homogenized as previously described [47] in RLT buffer supplemented with 1% β-mercaptoethanol. After RNA isolation, DNA digestion was performed by incubating 350 ng of RNA per sample with 2.5% DNase and 10% RDD buffer (QIAGEN, Hilden, Germany) for 30 min at room temperature. Reverse transcription was performed using the SuperScript IV cDNA synthesis kit (Thermo Fisher Scientific, Waltham, MA, USA). Gene expression was quantified using the SensiMix™ II Probe Master mix (Bioline, London, UK) on a StepOnePlus™ Real-Time PCR System (Applied Biosystems™, Thermo Fisher Scientific, Waltham, MA, USA) with primers, as listed in Appendix A.

The qPCR results were expressed as mean HEV-ORF3 vector genome copy number per µg total RNA (vg/µg). Known copy numbers of the HEV p6 plasmid were serially diluted and used to generate a standard curve.

### 2.5. Western Blot Analysis

Mouse tissue (10–20 mg) was lysed in 500 µL RIPA buffer (Thermo Fisher Scientific, Waltham, MA, USA) supplemented with 1 mM protease inhibitor (Roche, Basel, Switzerland). Tissue samples were homogenized as described above and centrifuged at 17,000× *g* for 15 min at 4 °C. Protein concentration was measured using the BCA Protein Assay Kit (Thermo Fisher Scientific, Waltham, MA, USA). Proteins were separated by 12% sodium dodecyl sulfate-polyacrylamide gel electrophoresis, followed by transfer onto a polyvinylidene fluoride membrane (EMD Millipore, Billerica, MA, USA), as previously described [20].

Western blot analysis was performed by using specific primary antibodies (rabbit anti-ORF3 antibody 1:1000 (a kind gift from Suzanne Emerson, NIH), mouse anti-GAPDH antibody 1:10,000 (Sigma-Aldrich, St. Louis, MO, USA), mouse anti-actin antibody 1:1000 (Sigma-Aldrich, St. Louis, MO, USA)), and secondary horseradish peroxidase (HRP)-conjugated antibodies (monoclonal goat-anti-rabbit or goat-anti-mouse antibody, 1:4000; Thermo Fisher Scientific, Waltham, MA, USA). Membranes were incubated with Pierce™ ECL Western Blotting Substrate (Thermo Fisher Scientific, Waltham, MA, USA) and analyzed using the ECL imager (ChemoStar, Intas, Göttingen, Germany).

For the detection of anti-ORF3 antibodies in mouse sera, cell lysates were obtained by transfection of HEK-293 cells with an ORF3-expressing plasmid (pscAAV-ORF3, described above) and a mock control, using a JetPrime transfection reagent (VWR, Radnor, PA, USA). Cells were harvested 48 h post-transfection. Lysis in RIPA buffer with protease inhibitor and centrifugation was performed as described above. Each cell lysate was loaded into one large well of a 12% SDS-PAGE gel. After blotting as described above, the membranes were cut into strips and incubated with the mouse serum diluted 1:100 in 5% milk in PBS at 4 °C overnight. Each primary antibody was used to stain an ORF3-protein containing membrane strip as well as the mock control membrane strip. The remaining steps were performed as described above. The membrane strips were realigned before imaging.

### 2.6. Production of eHEV Particles

HEV RNA was transcribed *in vitro* from the MluI-linearized Kernow-C1/p6 plasmid using the mMESSAGE mMACHINE™ T7 Transcription Kit (Thermo Fisher Scientific, Waltham, MA, USA) [48]. A total of 10 μg of *in vitro* transcribed viral RNA was electroporated into 4 × 10^6^ S10-3 cells using the Gene Pulser II apparatus (BioRad, Hercules, CA, USA) in 0.4 cm Gene Pulser cuvettes (BioRad, Hercules, CA, USA) at a capacity of 0.975 nF and a voltage of 0.27 V. On day seven post-electroporation, the cell culture supernatant containing eHEV particles was collected and filtered through a 0.45 µm pore size filter (Whatman GE Healthcare Life Sciences, Chicago, IL, USA).

### 2.7. Neutralisation of eHEV Particles

S10-3 cells were seeded at a density of 4x10^4^ cells per well into 48-well plates. Serum samples were diluted in DMEM and incubated with eHEV at a multiplicity of infection (MOI) of 1 × 10^−3^ at 37 °C for 60 min, before the solution was added to the cells. As controls, the following antibodies were used: polyclonal rabbit anti-ORF2 (a kind gift from Xiang- Jin Meng, Virginia Tech, USA) in a 1:200 dilution, polyclonal rabbit anti-ORF3 (a kind gift from Suzanne Emerson, NIH, USA) in a 1:200 dilution, monoclonal mouse anti-ORF2 antibodies 1E6 and 4B2 (Merck Millipore, Billerica, MA, USA) mixed 1:1 and then added in a 1:50 dilution, or recombinant mouse anti-ORF3 antibodies RB198 and RB200 (Geneva Antibody Facility, Geneva, Switzerland) mixed 1:1 and then added in a 1:50 dilution. Eight hours post-infection, the inoculum was removed and culture medium was replenished.

The cells were fixed in 4% paraformaldehyde (EMS, Hatfield, PA, USA) seven days post-infection and stained as described previously [49] using an anti-ORF2 monoclonal antibody (1:400, 1E6; Millipore, Burlington, MA, USA) and a secondary anti-mouse antibody conjugated to Alexa Fluor 594 (1:500; Thermo Fisher Scientific, Waltham, MA, USA). Foci-forming units (FFU) were quantified by counting ORF2-positive clusters.

## 3. Results

### 3.1. Generation of HEV ORF3-Expressing AAV Vectors and Administration to Mice

To express HEV ORF3 from an AAV vector, we cloned the ORF3 gene from the HEV GT3 Kernow-C1 p6 strain into the self-complementary pscAAV-CMV-EYFP-BGH polyA vector plasmid. In this construct, one of the AAV inverted terminal repeats (ITRs) is truncated, leading to a self-complementary genome configuration which circumvents the rate-limiting step of host-dependent double-strand (ds)DNA conversion of the AAV genome, and thus mediates faster and stronger transgene expression [28,42]. We additionally inserted an untranslated 450 bp fragment of the YFP sequence after the BGH polyA terminator to bring the length of the entire insert between the ITRs up to the optimal ~2000 bp. As a control, we used the YFP-expressing pscAAV-CMV-EYFP-BGH polyA vector plasmid described above [45].

These two recombinant AAV genomes were then encapsidated into the synthetic capsid AAVMYO3, which we have recently engineered in our lab (D.G.) in a semi-rational manner for high efficiency and specificity in the murine musculature following peripheral delivery (El Andari et al., submitted). After producing and purifying the AAV particles (Figure 1C), we titrated AAV genomes via RT-qPCR, and prepared dilutions of 1 × 10^11^ or 1 × 10^12^ AAV vg in PBS. Six-week-old female BALB/c mice were separated into two groups of four mice each. Before injection, we collected an initial blood sample. Then, we injected one group with 1 × 10^11^ ORF3 AAV vg and the other with 1 × 10^12^ ORF3 AAV vg through the tail vein (Figure 1D). Additional control groups comprising two mice each were injected with 1 × 10^11^ or 1 × 10^12^ YFP AAV vg, respectively.

Every two weeks, a blood sample was taken to monitor the antibody response. After 10 weeks, none of the mice showed any weight loss or signs of pathology. All mice were euthanised by terminal bleeding under intraperitoneal anaesthesia prior to the harvesting of serum, quadriceps femoris muscles, livers, and spleens for further analysis (Figure 1D).

### 3.2. AAV Dose-Dependent and Muscle-Specific Expression of HEV ORF3 in Mice

As shown in Figure 2, both RT-qPCR and Western blot (WB) analysis revealed ORF3 expression in the muscle, but not in the spleen or the liver of ORF3-transduced mice. In addition, we observed an AAV dose-dependent ORF3 expression. While only one of the four mice injected with 1 × 10^11^ ORF3 AAV vg showed modest ORF3 expression, all four mice injected with 1 × 10^12^ ORF3 AAV vg expressed ORF3, albeit at varying levels. Based on the RT-qPCR data, the threshold for detection of the ORF3 protein by Western blotting was ~3 × 10^4^ ORF3 mRNA copies per μg of total RNA. Using a polyclonal anti-rabbit ORF3 antibody, we only detected the palmitoylated ORF3 (~15 kDa) form [50] in the muscle. As loading controls for the WB, we detected different housekeeping genes, due to their differential expression in the different tissues tested. These results confirmed that the myotropic AAV capsid MYO3 used in our study specifically transduces the muscle of injected mice.

### 3.3. AAV Dose-Dependent Anti-ORF3 Antibody Induction over Time in Mice

To detect potential anti-ORF3 antibodies induced in the mice upon AAV transduction, we generated a positive control in the form of lysates from HEK-293 cells ectopically expressing ORF3 (Figure 3), and used the mouse sera as the primary antibody to detect the ORF3 protein. As shown in Figure 3A, all mice injected with 1 × 10^12^ ORF3 AAV vg developed anti-HEV-ORF3 antibodies, congruent with the transgene expression detected in the muscle (Figure 2). The single mouse from the low-dose group that showed modest ORF3 expression had also generated detectable anti-ORF3 antibodies. The mouse sera were able to detect both the unmodified (~11 kDa) and palmitoylated ORF3 (~15 kDa) form. As a control, we also incubated mock-transfected HEK-293 cell lysates and did not observe any specific bands upon incubation with the mouse sera (Appendix A). Altogether, we confirmed that systemic infusion of a myotropic AAV vector can induce ORF3 expression and consequently trigger the formation of anti-ORF3 antibodies in mice.

Next, we assessed the dynamics of the antibody induction in mice over time. To this end, we used lysates from HEK-293 ORF3 cells, as mentioned before. As shown in Figure 3B, mouse #2 injected with 1 × 10^11^ ORF3 AAV vg developed anti-ORF3 Abs after four weeks post-injection. In contrast, mouse #1 injected with the 10-fold higher dose of 1 × 10^12^ ORF3 AAV vg already showed anti-ORF3 Abs as early as two weeks post-injection. ORF3 antibody production reached a plateau at week six post-injection in both mice. Comparable kinetics were also observed in the other three mice in the 1 × 10^12^ vg cohort (Appendix A).

### 3.4. Moderate Neutralisation of eHEV Particles by Anti-ORF3 Antibodies

Subsequently, we tested whether the anti-ORF3 antibodies induced in the mice could neutralise eHEV particles in vitro. In cell culture, nHEV particles can be harvested intracellularly, while eHEV particles are harvested from the extracellular culture supernatant of HEV-infected cells [20]. However, a fraction of nHEV particles can be released into the supernatant due to cell death.

To estimate the extent of released contaminating nHEV particles, we mutated the start codon of ORF3 in the Kernow C1 p6 strain (ΔORF3 virus). In the absence of ORF3, eHEV particles cannot be secreted [15,27] and accumulate intracellularly. As shown in Figure 4A, ΔORF3 particles (grey bar) harvested in the culture supernatant led to roughly 10% of extracellular Kernow C1 p6 WT (wild-type) virus infection (black bar). Due to their envelope, eHEV particles are protected from anti-ORF2 antibodies [20]. When treating extracellular WT particles with monoclonal or polyclonal anti-ORF2 antibodies, they were neutralised up to 10%. This suggested that roughly 10% of all infectious particles in the culture supernatants were contaminating nHEV particles. In contrast, monoclonal or polyclonal anti-ORF3 antibodies neutralised extracellular WT particles up to 50% (Figure 4A).

Finally, we tested whether the serum from the high-dose ORF3 AAV-transduced mice is capable of reducing extracellular WT particle infectivity (Figure 4B). Compared to control serum from mice transduced with YFP AAV particles, the sera from all four ORF3 mice inhibited extracellular HEV WT infection, albeit at moderate and varying levels. Two of the four ORF3 mouse sera significantly inhibited extracellular WT particle infection up to 50% compared to the YFP serum control when applied at a high concentration. We did not observe a clear correlation between the amount of induced ORF3 antibodies (Figure 3A) and their inhibitory capacity (Figure 4B).

## 4. Discussion

Following the first report in 1997 by Manning and colleagues, who showed that AAV-mediated expression of Herpes simplex glycoproteins in mice induced a potent cellular and humoral immune response [51], a number of studies also proposed AAV-based vaccine strategies against the Dengue virus [52], hepatitis C virus [53], and SARS-CoV-2 [38,54]. In addition, clinical phase I [55] and II [56] studies with an AAV-based vaccine against the human immunodeficiency virus showed a very good safety and tolerance profile. Based on these results, we were encouraged to assess the capacity of AAV as a vector-based vaccine candidate against HEV. Specifically, in view of the recent discovery of the quasi-enveloped form of HEV particles, we sought to induce antibodies against ORF3 and to re-evaluate their neutralisation capacity.

To this end, we used the novel chimeric myotropic AAVMYO3 capsid that we have recently developed in our (D.G.) laboratory (El Andari et al., submitted) to express ORF3 in the musculature of mice from a minimally invasive, peripheral tail vein injection. We found that the ORF3 transgene was specifically expressed in the muscle tissue of AAV transduced mice, but not in their spleen or liver (Figure 2). The pronounced detargeting from the liver mediated by the AAVMYO3 variant is a seminal benefit for a vaccine strategy based on in vivo antigen expression, as liver-directed gene therapy can induce systemic tolerance to the delivered transgene [57].

As shown in Figure 2 and Figure 3, a higher amount of injected AAVs led to higher tissue expression of the ORF3 transgene, on both the mRNA and protein level. We also measured a good correlation between the ORF3 transgene expression and the induction of anti-ORF3 antibodies. We observed a detectable antibody induction against ORF3 as early as two weeks, which plateaued at roughly six weeks post-injection. While we could not resolve the possible underlying mechanisms in this proof-of-concept work, it will be informative in follow-up studies to harvest the muscle tissues at earlier time points and study whether ORF3 expression levels also plateau or even decline over time, such as due to promoter silencing or epigenetic inactivation of the AAV genomes. Regardless of mechanism, these results suggest that a booster shot could be useful to further enhance the immune response against ORF3. To this end, it will be pivotal to swap the capsid of the AAV vector used for the booster from AAVMYO3 to another variant that will not or will only be poorly neutralised by the anti-AAVMYO3 antibodies induced by the prime vaccine. Luckily, in principle, this is feasible using AAV capsid evolution and pseudotyping technology [28,30,32,33,34,35].

As the vaccine target in our study, we chose ORF3, a small 113 aa phosphorylated protein that is critical for eHEV secretion [26,27]. A recent study showed that ORF3 is present on the cytosolic side of the quasi-envelope [50]. However, ORF3 has been shown to have a viroporin function [58], suggesting that some ORF3 epitopes may be exposed on eHEV particles. This is supported by an earlier study showing that anti-ORF3 mAbs are able to capture viral particles from supernatants of HEV-infected cultured cells and serum samples of HEV patients, but not from their faeces [24]. In the same study, the authors demonstrated that an anti-ORF3 mAb was able to partially neutralise eHEV in vitro. Additional studies showed that immunocapture of eHEV by anti-ORF3 mAbs was possible, albeit at low efficiency [15,26]. In light of these data, it seems possible that at least some ORF3 epitopes are accessible on the surface of the quasi-enveloped particles.

In our study using either characterised antibodies or the sera from ORF3-transduced mice, we observed a moderate neutralisation effect of anti-ORF3 antibodies on extracellular, supposedly eHEV particles. While our results agree with previous work, in which the authors observed a delayed onset of putative eHEV particle infection following a prior treatment with anti-ORF3 antibodies [24], they remain difficult to translate to in vivo use of ORF3 as a vaccine candidate, since a protective effect relies on both the cellular and the humoral immune response. We also noted a discrepancy between the amount of antibodies in the mouse sera and their eHEV neutralisation capacity, especially for mouse #1 in the AAV-ORF3 1 × 10^12^ vg group. Possible explanations include the induction of antibodies that bound but did not neutralise ORF3, perhaps owing to their epitopes being located in the cytoplasmic and/or transmembrane ORF3 regions, which are likely inaccessible on eHEV particles. Epitope mapping of the induced ORF3 antibodies could help to dissect these phenotypes, and should therefore motivate follow-up studies.

To date, only two studies have evaluated ORF3 as a vaccine target to prevent HEV infections in primates and chickens, respectively [59,60]. These studies showed that immunisation with ORF3 led to partial protection against HEV infection, since fewer but not all animals were infected and those that were infected had a shortened duration of viremia and milder disease symptoms. However, in both studies, naked, faeces-derived HEV (nHEV) was used and applied intravenously for the challenge (as reviewed in [61]). These vaccines may have prevented or limited the spread of progeny eHEV particles in infected animals, which then may have either cleared the infection directly or developed only mild disease. The neutralisation of eHEV particles by anti-ORF3 antibodies demonstrated in our own work supports this hypothesis and should motivate additional in vivo vaccination studies.

## 5. Conclusions

Our results imply that systemic delivery of our novel muscle-tropic and self-complementary AAVMYO3 vector from a minimally invasive route could be a potent strategy to induce ORF3 expression and the generation of anti-ORF3 antibodies in vivo. Concurrently, our pilot study outlines additional preclinical experiments and analyses that should be conducted to enable a possible clinical translation of our concept and vector, such as a comparison of various routes of administration, especially intravenous versus intramuscular AAV injection. In addition, it will be interesting and informative to study whether the combination with booster shots will further enhance antibody induction, and consequently, eHEV neutralisation capacity.

Clearly, HEV ORF3 remains an interesting vaccine target owing to recent observations of HEV transmission through blood transfusions [9] and the possible role of eHEV particles in the dissemination to other tissues in chronic patients [22]. While we observed a moderate neutralising effect of anti-ORF3 antibodies in vitro in our proof-of-concept study, an in vivo challenge following an optimized AAV-ORF3 vaccination regime may yield more compelling effects and should thus be an interesting task for future work.

Finally, we point out the encouraging fact that, in principle, the platform presented here can also be harnessed for the quick and cost-effective evaluation of other viral proteins as vaccine targets. A particularly interesting candidate might be HEV ORF4, for which a role in the enhanced mortality observed in HEV GT1-infected pregnant women has been suggested recently [62].

## Figures and Tables

**Figure 1 viruses-14-00266-f001:**
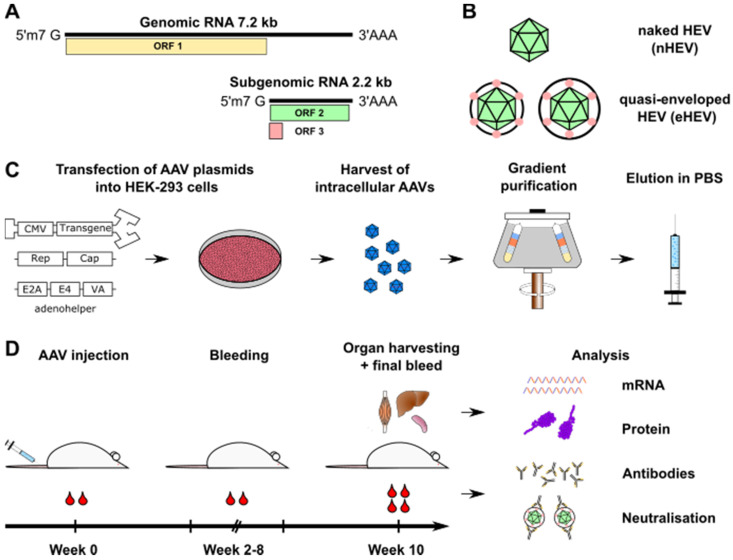
Experimental overview. (**A**) The full-length 7.2 kb HEV genome encodes the replicase ORF1, and a 2.2 kb subgenomic HEV RNA encodes capsid ORF2 and the small phosphoprotein ORF3 [43]. (**B**) HEV exists in two forms: a non-enveloped virion (nHEV) found in faeces/intracellularly and a quasi-enveloped virion (eHEV) found in the blood stream and cell culture supernatant. Whether ORF3 is (partially) presented on the outside of eHEV particles remains unclear. (**C**) Production of AAV vectors was accomplished by triple-transfection of HEK-293 cells with a recombinant AAV vector plasmid encoding ORF3 or YFP (yellow fluorescent protein), an AAV helper plasmid encoding *rep* of AAV2 together with the synthetic capsid gene MYO3, as well as an adenoviral helper plasmid. This was followed by a harvest of the AAVs from the cell pellet and their purification by iodixanol density gradient centrifugation, buffer exchange to PBS, titration of AAV genome copies, and particle concentration. (**D**) Initial blood (60–80 μL) was collected from the facial vein before intravenous injection of 1 × 10^11^ or 1 × 10^12^ AAV vector particles encoding ORF3 or the YFP control (always diluted in 100 μL PBS) into four mice each. Mice were bled every two weeks (60–80 μL) to monitor the antibody response. At week 10, mice were euthanised by a final bleed under intraperitoneal anaesthesia. Their quadriceps femoris muscle, liver, and spleen were harvested for further analysis, as indicated.

**Figure 2 viruses-14-00266-f002:**
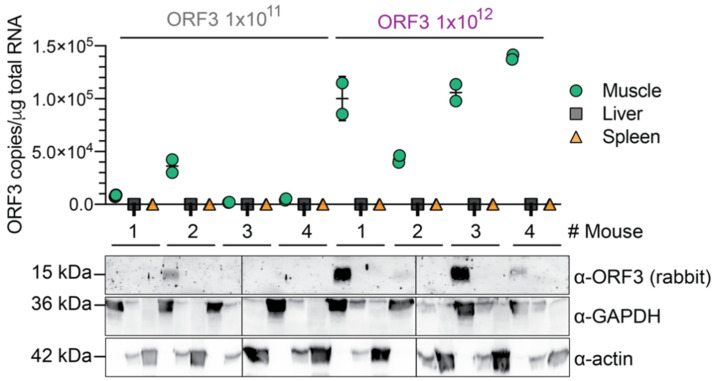
Muscle-specific expression of HEV-ORF3 in AAVMYO3-transduced mice. HEV ORF3 copy numbers were quantified via RT-qPCR in the quadriceps femoris muscle (green), liver (grey), and spleen (orange) of mice transduced with the indicated amount of vg ORF3 AAVs. Results represent the mean of *n* = 2 ± SD. ORF3 protein expression was analysed in respective mouse tissues via Western blotting. ORF3 expression was detected using a rabbit anti-ORF3 polyclonal Ab, as well as housekeeping genes GADPH or actin using respective mAbs. Palmitoylated ORF3 = ~15 kDa, GAPDH = ~36 kDa, actin = ~42 kDa.

**Figure 3 viruses-14-00266-f003:**
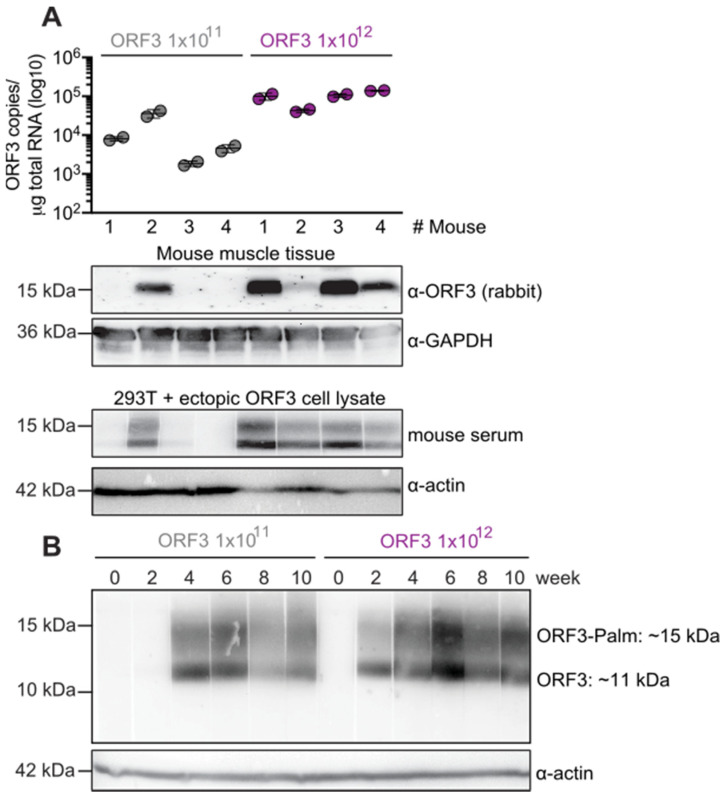
Induction of anti-ORF3 antibodies correlates with ORF3 expression levels in mice. (**A**) Correlation of ORF3 copy numbers (data from Figure 2) and ORF3 protein expression in the muscle with anti-ORF3 antibodies in the serum of ORF3 AAV-transduced mice. ORF3 expression in the muscle was detected using the rabbit anti-ORF3 polyclonal Ab described above. To visualise the anti-ORF3 antibodies, HEK-293 cells transfected to express ORF3 were lysed and loaded onto a 12% SDS-PAGE. After separation and transfer onto a PVDF membrane, ectopic ORF3 was detected in the presence of anti-ORF3 antibodies in the mouse sera and visualised by incubation with anti-mouse HRP-conjugated Abs. As a loading control, actin was detected using an anti-actin mAb. (**B**) Dynamics of antibody induction in mice over 10 weeks. HEK-293 cells transfected to express ORF3 were lysed and loaded onto a 12% SDS-PAGE. After separation and transfer, the membrane was cut into individual pieces and incubated with sera from mouse #2 (1 × 10^11^ ORF3 AAV) or mouse #1 (1 × 10^12^ ORF3 AAV) harvested at the indicated different time points post-AAV injection, followed by a secondary anti-mouse antibody. As a loading control, actin was detected using an anti-actin mAb. ORF3: = ~11 kDa, palmitoylated ORF3 = ~15 kDa, GAPDH = ~36 kDa, actin = ~42 kDa.

**Figure 4 viruses-14-00266-f004:**
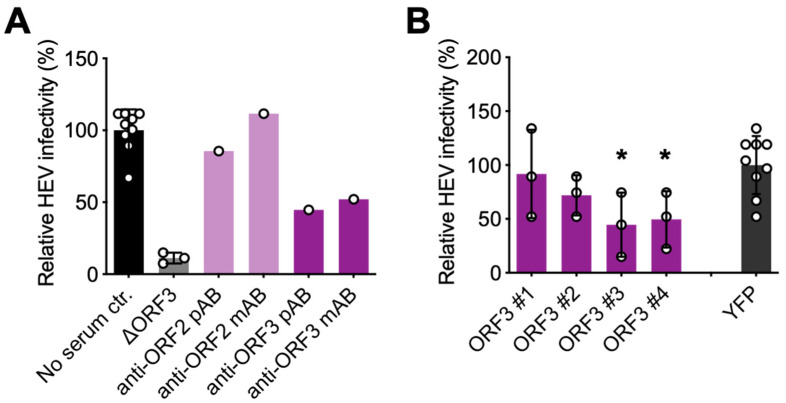
Inhibition of extracellular HEV particles by anti-ORF3 antibodies and sera from ORF3 AAV-transduced mice. (**A**) Extracellular HEV particles harvested from hepatoma S10-3 cells electroporated with in vitro transcribed HEV RNA were incubated with the indicated antibodies for 1 h prior to infection of S10-3 cells (anti-ORF2 and ORF3 pAb 1:200, anti-ORF2 mAb mix 1:50, anti-ORF3 mAb mix 1:50). HEV infection was normalised to untreated extracellular HEV particles (black bar). To assess the background of nHEV particles, supernatant from S10-3 cells electroporated with ΔORF3 virus was also titrated (grey bar). (**B**) Extracellular HEV particles were incubated with a 1:50 dilution of serum from mice harvested 10 weeks post-ORF3 AAV transduction and titered on S10-3 cells. HEV infection was normalised to extracellular HEV particles incubated with mouse serum transduced with YFP (dark grey bar). HEV infections were quantified by staining ORF2-positive cells 7 days post-infection when HEV replication in cultured cells peaks. Results represent the mean of *n* = 3 ± SD. Statistical analysis was performed using an unpaired, two-tailed t-test with *: *p* < 0.05.

## Data Availability

Not applicable.

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
