# Peer review of "Induction of Hepatitis E Virus Anti-ORF3 Antibodies from Systemic Administration of a Muscle-Specific Adeno-Associated Virus (AAV) Vector"

_viruses, 2022, doi:10.3390/v14020266_

Round 1

Reviewer 1 Report

The authors in this study have developed a platform for vector-based vaccine for HEV. The vector developed is Adeno-Associated Virus Vector (synthetic and myotropic). For the present study ORF3 of HEV was used for the vector based vaccine and this induced in vivo (mice) anti-ORF3 which had modest neutralization properties.

I have following comments to make:

  1. In introduction authors gave a good account of the HEV family and went ahead to describe the significance of three ORF’s namely ORF1, ORF2 and ORF3. Another ORF namely ORF4 in HEVgt1 has come up of significance in the pathogenesis of HEV in pregnant women as this protein is involved in enhancing the HEVgt1 replication (Viruses202113(7), 1329). A mention of this here is important as incorporation of ORF4 in the suggested platform may be exploited in future to reduce HEVgt1 associated mortality in pregnant women.
  2. Authors have defined the two forms of the virus naked nonenveloped ( nHEV) in tissues and enveloped (eHEV) in blood. In the later envelope contains ORF3. Authors discuss that the eHEV protects the virus from anticapsid antibodies generated from the 2 vaccines, which have undergone trials in Nepal and China. The latter is commercially available as of today in China and few select countries outside China. In spite of what authors mentioned both vaccines have been highly efficacious and role of anticapsid antibodies in tissues (nHEV) may be and I believe more important. Authors should address this issue in their statements.
  3. Authors should comment why they choose to use ORF3 for this vector-based vaccine and not ORF2 as done for 2 earlier vaccine. The explanation that eHEV is not neutralized by antibodies because of the envelope is not as valid as authors think as such vaccines have shown to be effective.
  4. Is it possible to incorporate ORF4 in this vector platform and exploit it to reduce mortality in pregnant women?

Reviewer 2 Report

In the present manuscript, the authors have characterized a myotropic adeno-associated virus vector (AAVMYo3) to express the small HEV phosphoprotein (ORF3) (found on quasi-enveloped HEV) in the musculature of mice and observed the dose-dependent formation of anti-ORF3 antibodies. The serum of ORF3 AAV-transduced mice showed a modest but varying inhibitory effect on the infection of quasi-enveloped HEV in cell culture.

  The study does not report novel findings on HEV biology but describes a useful tool that is of particular interest for the filed and could contribute in the future to the development of vector-based vaccines against HEV and other infectious agents.

Comments:

  1. 3B (ORF3 1x1012): the result only for mouse #1 is provided. To confirm the reproducibility and obtain plausible data, the results from three other mice (#2, #3 and #4) should also be shown.
  2. 4B: Inhibitory effects were at moderate and varying levels and no clear correlation between the amount of induced ORF3 antibodies (Fig. 3A) and their inhibitory capacity (Fig. 4B) was observed. The authors are encouraged to access the inhibitory effects at different time points (other than 7 days post-infection) to confirm the inhibitory effects and should discuss the reason for varying inhibitory effects among sera obtained from four ORF3 mice tested.
  3. Typos are scattered throughout the manuscript. For example, SARS-CoV2 (line 109), six-weeks-old (lines 164 and 253), ul (line 187), mins (line 189), um (line 219), and SARS CoV2 (line 372).
